# Field experiments to assess passage of juvenile salmonids across beaver dams during low flow conditions in a tributary to the Klamath River, California, USA

**Michael M. Pollock**[1]*, **Shari Witmore**[2], **Erich Yokel**[3]

**1** National Oceanic and Atmospheric Administration, Northwest Fisheries Science Center, Watershed Program, Seattle, Washington, United States of America, **2** National Oceanic and Atmospheric Administration, National Marine Fisheries Service, West Coast Region, Klamath Branch, Arcata, California, United States of America, **3** Scott River Watershed Council, Etna, California, United States of America

* michael.pollock@noaa.gov

**Data Availability Statement:** All relevant data are within the paper and its Supporting Information files.

## Abstract

Across Eurasia and North America, beaver (*Castor* spp), their dams and their human-built analogues are becoming increasingly common restoration tools to facilitate recovery of streams and wetlands, providing a natural and cost-effective means of restoring dynamic fluvial ecosystems. Although the use of beaver ponds by numerous fish and wildlife species is well documented, debate continues as to the benefits of beaver dams, primarily because dams are perceived as barriers to fish movement, particularly migratory species such as salmonids. In this study, through a series of field experiments, we tested the ability of juvenile salmonids to cross constructed beaver dams (aka beaver dam analogues). Two species, coho salmon (*Oncorhynchus kisutch*) and steelhead trout (*O. mykiss*), were tracked using passive integrated transponder tags (PIT tags) as they crossed constructed beaver dam analogues. We found that when we tagged and moved these fishes from immediately upstream of the dams to immediately downstream of them, most were detected upstream within 36 hours of displacement. By the end of a 21-day field experiment, 91% of the displaced juvenile coho and 54% of the juvenile steelhead trout were detected on antennas upstream of the dams. In contrast, during the final week of the 21-day experiment, just 1 of 158 coho salmon and 6 of 40 (15%) of the steelhead trout were still detected on antennas in the release pool below the dams. A similar but shorter 4-day pilot experiment with only steelhead trout produced similar results. In contrast, in a non-displacement experiment, juveniles of both species that were captured, tagged and released in a pool 50 m below the dams showed little inclination to move upstream. Further, by measuring hydraulic conditions at the major flowpaths over and around the dams, we provide insight into low-flow conditions under which juvenile salmonids are able to cross these constructed beaver dams, and that multiple types of flowpaths may be beneficial towards assisting fish movement past instream restoration structures. Finally, we compared estimates of the number of juvenile salmonids using the pond habitat upstream of the dam relative to the number that the dam may have prevented from moving upstream. Upstream of the dams we found an abundance of juvenile

**Funding:** The authors received no specific funding for this work.

**Competing interests:** The authors have declared that no competing interests exist.

salmonids and a several orders of magnitude difference in favor of the number of juveniles using the pond habitat upstream of the dam. In sum, our study suggests beaver dams, BDAs, and other channel spanning habitat features should be preserved and restored rather than removed as perceived obstructions to fish passage.

## Introduction

Human-constructed dams and other instream obstructions have become a ubiquitous feature across riverine landscapes and have altered many natural processes by reducing ecosystem connectivity. In the past five millennia, millions of dams have been constructed by humans, with over two million built in the USA alone [1, 2]. Currently, efforts are underway to remove many of these dams, with the primary objective of restoring stream connectivity, and more specifically, to improve fish passage [3, 4].

While the number of dams built by humans is impressive, there are actually fewer dams in North America now than prior to European colonization, albeit of a different size and materials. Historic estimates of North American beaver (*Castor canadensis*) populations range from 60–400 million, suggesting that across their $1.5 \times 10^7$ km$^2$ range, there was anywhere from 10–60 million beaver dams, mostly made of sticks and mud [5–7]. In addition, large wood formed millions of jams, dams and other obstructions that dammed and diverted sediment and water across streams, rivers and even entire valleys [8–10]. Historic accounts support the ubiquity of such biogenic dams. Walter and Merritts (2008) [11] in their comprehensive study of pre-European paleo-channels along mid-Atlantic seaboard of eastern North America, determined that many were so heavily impacted by beaver dams and vegetation, that there were few discernable channels. This description is consistent with the early depictions of valley bottoms as ubiquitous swampy meadows and marshes. On the other side of the continent, the Willamette Valley (13,700 km$^2$) in Oregon was described by some of the first Europeans to see it (e.g. beaver trappers) as full of wood jams and rafts that created ever shifting multiple channels and backwaters and extensive marshes across the valley, such that travel was limited to trails on edges [12]. Similarly, at our study site on the Scott River (area = $2.1 \times 10^3$ km$^2$) a major tributary to the Klamath River in California, the valley floor was described by trappers as "all one swamp, caused by the beaver dams, and full of (beaver) huts" [13].

Through commercial trapping for furs, and government-sponsored desnagging, stream cleaning and wildlife control, humans have removed most of these biogenic dams, jams and other obstructions, and most of this occurred prior to the 20th century [10, 14]. Many scientific disciplines related to the study of rivers such as ecology, geology, and fluvial geomorphology emerged in the late 19th and early 20th centuries, and subsequent to the widespread removal of these obstructions to flow and sediment transport., This has profoundly influenced the perception among scientists and natural resource managers even to this day that the natural and ideal condition of all streams is "free-flowing" and clear of dams and other obstructions [2, 10, 15].

However, such biogenic, wood-based dams are fundamentally different from modern concrete and rock dams in that they are small (very low-head), semi-permeable and ephemeral. Beaver dams in particular are usually small, not exceeding 2 m in height (mostly < 1 m high), and are transitory landscape features, with dam lives typically ranging from a few years to decades [14, 16–18]. Such dams have enormous beneficial ecosystem impacts, such as creating ponds, wetlands and other types of slow-water habitat, contributing to water storage and groundwater recharge across landscapes, altering sediment transport rates and stream morphology and changing the underlying geomorphic structure across entire valley floors [19–25].

Thus, throughout much of the northern hemisphere, beaver have been creating structurally complex and biologically diverse aquatic habitat for millions of years, and many anadromous and freshwater fishes have adapted to and evolved in such habitat [26, 27]. In addition to dams, beaver create complex habitat through the construction of lodges and caches made of wood from nearby trees that they fell, as well as the excavation of soil to build canals, channels, tunnels and burrows [5]. Such activities create an aquatic environment that is biologically, hydraulically, thermally and structurally diverse.

In North America, over 80 fishes are known to use beaver ponds, with 48 species commonly using them, inclusive of commercially, culturally and recreationally important species such as coho salmon (*Oncorhynchus kisutch*), steelhead trout (*O. mykiss*), Atlantic salmon (*Salmo salar*), cutthroat trout (*O. clarkii*) and brook trout (*Salvelinus fontinalus*) [6]. Many fishes use the structurally complex, deep, slow water and emergent wetlands created upstream of beaver dams [26, 28]. Beaver build dams typically ranging from 30–100 cm, but may be as high as 250 cm, and the height of such dams has raised concerns that they are barriers to fish passage, particularly for salmon and trout [28, 29]. In the United States, state and federal rules often require stream passage barriers to be no more than 15–20 cm in height, making most natural beaver dams non-conforming to existing guidelines [30, 31]. There are also concerns about steep stream gradients as fish passage barriers, and typically when constructing passage routes over barriers such as dams, a series of step-pools is created rather than a steep stream bed. Such rules are in place to ensure that human-built structures such as culverts, hydroelectric, water storage and diversion dams do not obstruct the natural movement of fishes. Globally, the rapid increase in large dam construction highlights the need to understand migratory behavior and passage needs for many fishes [28], and much effort has gone into designing and carefully engineering constructed fishways that ideally allow for fish passage over such structures [31].

At the same time, in Europe and North America natural resource policy guidance documents intended to facilitate recovery of fish and wildlife populations stress the need for more channel-spanning instream restoration structures such as beaver dam analogues (BDAs), log steps, boulder weirs, log jams and natural beaver dams, to create dynamic, structurally complex and spatially diverse aquatic, riparian and wetland habitat [32–37].

Fish passage rules designed for large dams, culverts and other obstructions are typically applied to restoration structures, even though their scale, purpose and function is quite different. In particular, restoration structures designed to be analogous to beaver dams (BDAs) in both form and function, are becoming an increasingly popular stream restoration technique [38–44].

In this study, we assess how salmonid species navigate past a beaver dam analogue constructed as part of a restoration project to help recover the Endangered Species Act (ESA)-listed Southern Oregon-Northern California Coast population of coho salmon [45]. The primary objective of this study was to evaluate whether juvenile coho salmon can pass upstream over beaver dam analogues and if so, identify a preferred flow path, e.g., do they prefer to jump over or swim around the BDAs? Steelhead trout are also a native species in the study area that use the beaver ponds, though to a lesser extent than coho salmon, so we also evaluated their ability to pass upstream over these structures.

Our second objective was to provide a basis for making a comparison between the number of fish that benefited from the habitat created upstream of the BDA and the number that may have been prevented from moving upstream because of the BDA. We hypothesized that, because these salmonids have evolved in the presence of beaver dams for millions of years, that they have also evolved strategies for crossing them, and that by constructing dams similar to beaver dams in terms of size, location and materials, these fishes would also be able to cross

these human-built structures. The results of this study are intended to guide future design considerations for fish passage at stream restoration structures.

## Site description

The study took place in northern California on Sugar Creek, a tributary to the Scott River, which is itself a major tributary to the Klamath River (Fig 1). The Scott River watershed (HUC #18010208) encompasses 2,105 km$^2$ and is located in the Klamath and Marble Mountains of Western Siskiyou County in Northwest California (Fig 1).

When European trappers first arrived in the 1830s, the valley floor of the Scott River was so full of beaver dams and lodges that it was in essence one large wetland [13]. Because of this abundance, it was initially called the Beaver Valley, and trappers rapidly removed thousands of beavers [13, 46]. Today a small number of beaver persist in the watershed in a few streams, including Sugar Creek. The area also has a history of extensive gold mining and the study reach on Sugar Creek is in an area that has been dredged for gold as recently as the mid-twentieth century, and currently flows through large mounds of cobble-dominated mine tailings.

The bedrock in the area, dating from pre-Silurian to Late Jurassic and possibly Early Cretaceous time, consists of consolidated rocks whose fractures yield water to springs at the valley margins and in the surrounding upland areas [47]. The valley alluvial fill consists of a few isolated patches of older alluvium (Pleistocene) found along the valley margins and of younger alluvium which includes stream-channel, floodplain, and alluvial-fan deposits of recent age [47]. Recent alluvial deposits reach a maximum of more than 120 m thick in the wide central part of the valley. The average seasonal precipitation is 805 mm but may exceed 1780 mm annually in the western mountains, and exceed 760 mm in the eastern mountains. The average annual temperature in the valley is 10.2˚C. Streamflow in the Scott River is primarily driven by annual fluctuations in snowpack. Most of the watershed is forested with conifers, predominantly Ponderosa pine (*Pinus ponderosa*) and Douglas-fir (*Pseudotsuga menziesii*), transitioning into oak (*Quercus* spp) savannah on the lower foothills, and then to pasture and irrigated fields on the main valley floor, with cottonwood (*Populus trichocarpa*) and willow (*Salix* spp) lining the major streams in narrow bands between the channel and the frequently rip-rapped banks.

## Methods

As part of an experimental stream restoration project intended to improve habitat for ESA-listed coho salmon, in 2015 we constructed two BDAs on Sugar Creek (UTM 10T 514732E, 4576739N) approximately 50 m and 200 m above its confluence with the Scott River, following the methods as described in [40, 48]. Such structures are intended to mimic the form and function of beaver dams, and under ideal conditions, they are eventually colonized by beaver. The structures were made by pounding a line of posts into the ground, approximately perpendicular to the direction of flow, then weaving willow between the posts. A downstream apron of cobbles was provided to minimize scour and an upstream berm of clay, organic material, sand and rock was constructed to create a semi-permeable structure with flow moving through, over and around the structure during most of the year, but with some side channel and side passage flow diminishing in the summer when flows decrease due to both natural causes and upstream water diversions. Although juvenile fish could likely moved through some of the pores within the structure, most of the flow was either over or around the structure and we thought that most fish would follow one of these major flow paths to cross the structures.

The lower BDA was constructed at the same location and height (approximately 1 m) as a naturally occurring beaver dam that had been abandoned a few years previous, and had a total

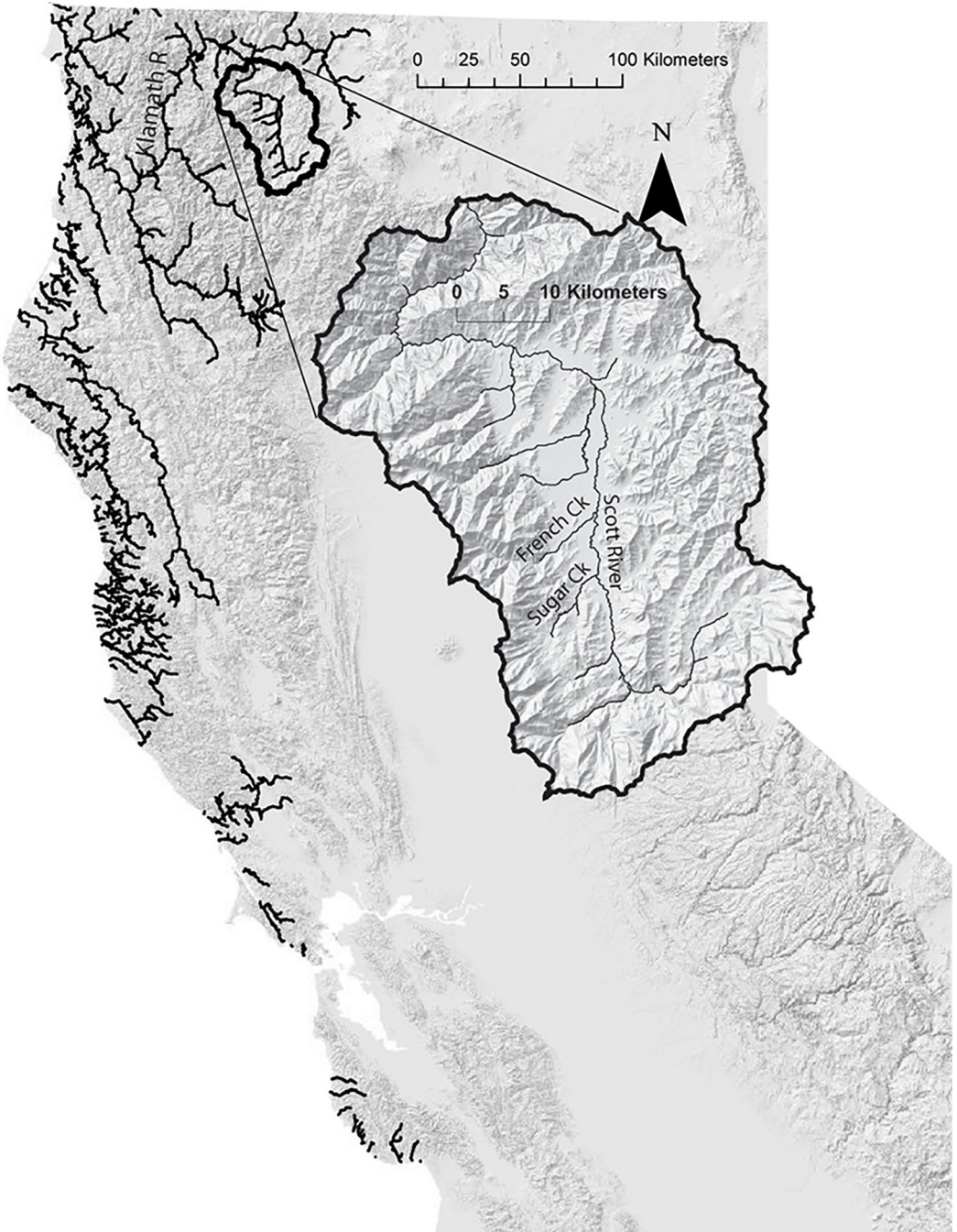

**Fig 1. Site map of the Scott River, a tributary to the Klamath River, California, USA.** Black lines indicate the current extent of coho salmon in California. Inset shows the topography of the Scott River watershed and the location of Sugar Creek in the upper watershed. California is located on the western coast of the United States of America, north of Mexico and south of the state of Oregon.

linear width of 45 m. The upper BDA was constructed in a relatively constricted reach between piles of mine tailing cobbles. The crest elevation was approximately 30 cm above the downstream pool created by the lower BDA, and the total width was 15 m. In the summer of 2017, two smaller BDAs were constructed downstream of the lower BDA to provide additional stability of the structure and to address perceptions that the 1 m-high structure was a barrier to

fish passage. As of summer, 2017, the BDAs had created approximately 7100 m$^2$ of slow water habitat and wetlands, and were actively being colonized by beavers. Coho salmon have annually spawned in Sugar Creek above the BDAs, and the ponds have supported juvenile coho salmon and steelhead trout throughout the year.

### Experimental captures and releases

To assess the ability of juvenile salmonids to pass over the dams, we performed a series of three experiments in 2016 and 2017 by tagging and then displacing fishes from above to below a dam or dams, or by tagging fishes below dams and then monitoring to see if they moved upstream. The obstruction of downstream movement was not perceived as a potential problem and was not monitored.

Fish to be tagged were captured with a two-person hand seine with (1/4") mesh cap (H. Christensen Co.), transferred to buckets equipped with battery-operated aeration pumps and brought to an onshore work station where they were anesthetized in small batches using alka-seltzer tablets. A 12 mm full duplex Passive Integrated Transponder (PIT) tag that resonates at 134.2 kHz (Biomark, Inc.) was inserted into the peritoneal cavity of each fish using a Biomark, Inc. syringe with disposable tips that was specifically designed for PIT tag insertion. Minimum allowable size of taggable salmonids was 65 mm, and almost all tagged fish were between 65–80 mm in length. Tagged fish were placed into a holding pen or bucket until they fully recovered (usually < 60 minutes), and then released.

**Experiment 1.** As a pilot study to assess whether juvenile salmonids could cross BDAs, on September 30, 2016, we captured and PIT-tagged 32 juvenile *O. mykiss* and placed them in the pool below the single BDA that was installed at that time (Fig 2). The following day we captured and tagged another 16 *O. mykiss* juveniles and released them in the same pool.

Fish to be tagged were captured with a two-person hand seine with (1/8") mesh cap (H. Christensen Co.).

A series of temporary, 60 cm by 60 cm square portable PIT antennas attached to a Biomark RM301 reader board with a multiplexor that "sampled" each antenna for 100 mS every 900 mS, were placed in the release pool, just above the BDA in the pond (Pond 1), and downstream of the block net, to monitor the movement of tagged fish and maximize the potential for detecting any fish that moved upstream past the BDA and into Pond 1 (Fig 2). Our arrays were not set up to detect fish that passed the dam by moving through the diffuse flow within the pores of the structure. PIT antennas were set up so that they covered approximately 90% of the total side channel area through which the fish could pass, and included the thalweg, which we assumed to be the most commonly used passage route.

For statistical analysis for all experiments, we used a binomial distribution to test for two possible outcomes, whether or not a fish was detected at a specific route crossing past a BDA. We describe the proportion of the fish that crossed an obstacle relative to the total sample size, and provide 95% confidence intervals.

There were numerous flow paths over, through and around the BDA, but the major flow path was a side channel that skirted the edge of the BDA on river left, with a discharge of approximately 0.03 m$^3$/s (about 1 cubic foot per second) or about half the total estimated discharge measured at a gage station approximately 1 km upstream (CA Dept. of Water Resources gage #F25890) at the time of the study (Fig 3). The side channel flowed over cobble and gravel for a distance of 8.3 m at a 10% slope, until it entered the pool immediately below the BDA.

Coho outmigration occurs from March through May and is typically centered around peak flow events. Vertical arrows indicate when mark and release experiments to test for passage

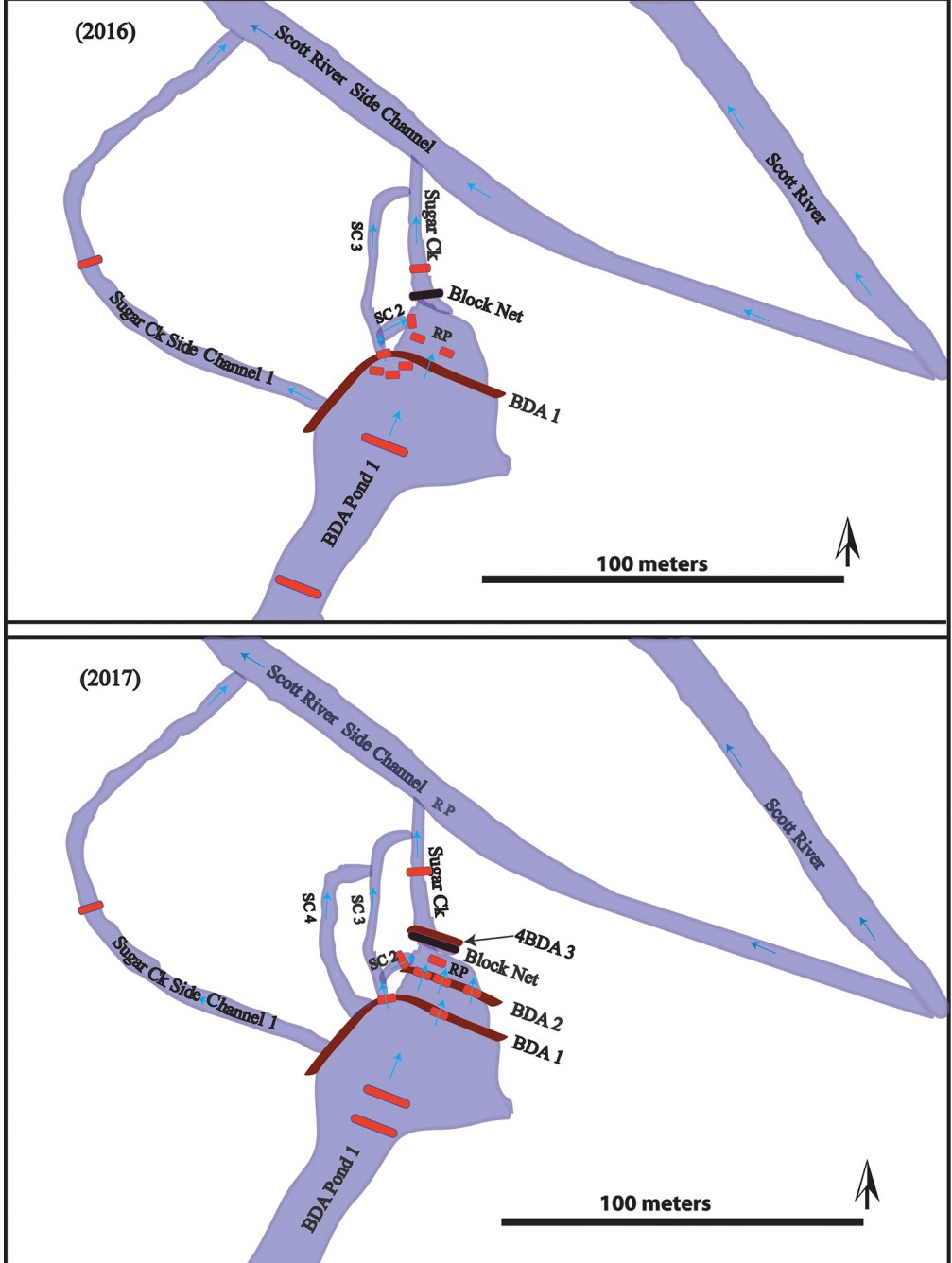

**Fig 2. Planview of fish passage experimental setups in 2016 and 2017.** The short red lines around BDAs indicate temporary PIT antennas, the long red lines indicate permanent PIT antennas; SC = side channel; RP = Release Pool, where tagged fish were released; BDA = Beaver Dam Analogue. Major flow paths are shown with blue arrows, though many minor flow paths exist throughout and between the BDAs. Blue shaded areas are places of BDA-influenced inundation.

across BDAs occurred, which were all during low flow periods at the end of September-early October, 2016, late October-early November, 2017, and July, August and September, 2017.

**Experiment 2.** During the summer and fall of 2017, we tagged 1,078 juvenile coho and 363 juvenile steelhead trout in the Sugar Creek beaver ponds with 12 mm full duplex PIT tags.

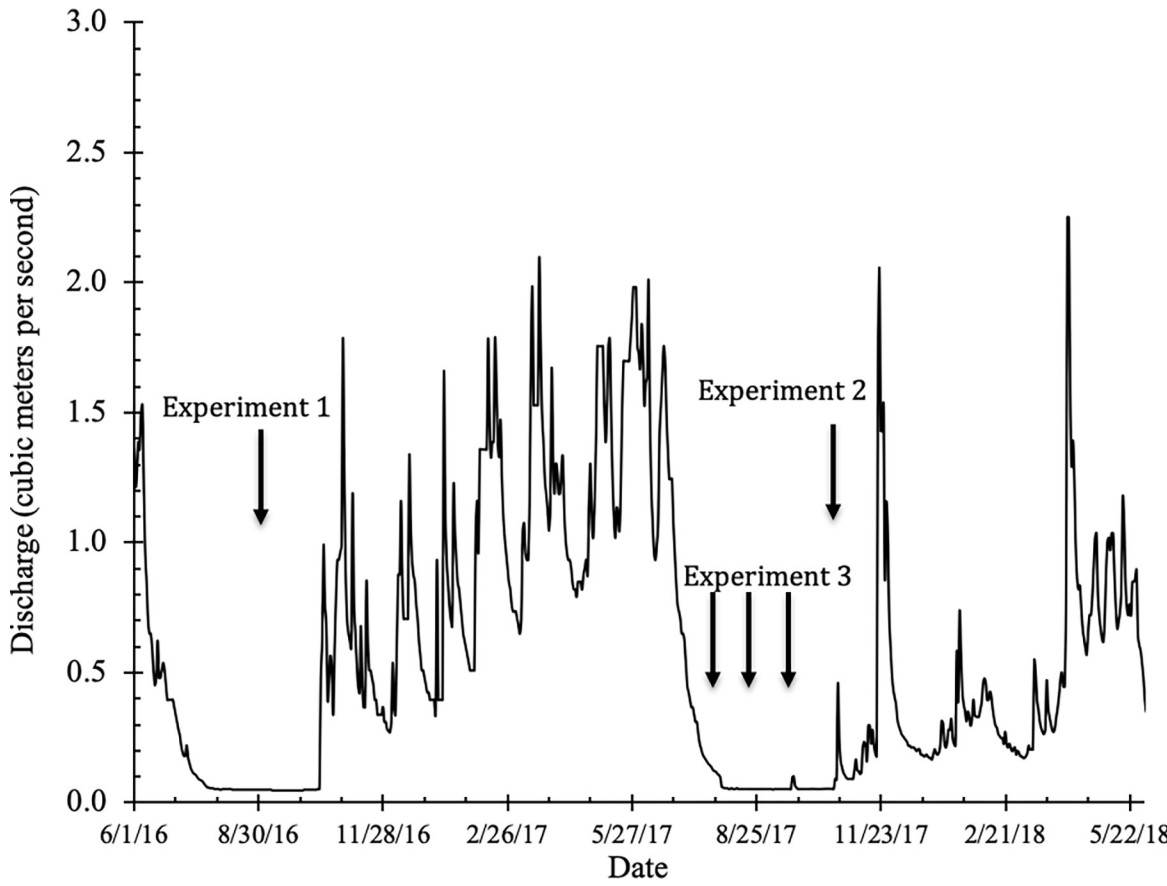

**Fig 3. Hydrograph of Sugar Creek during the period of study, showing the low flow conditions during which the studies took place, about 0.03 m³/s.** For comparison, the mean annual discharge of Sugar Creek is 0.5 m³/s.

We also opportunistically tagged 16 juvenile Chinook salmon (*O. Tshawytscha*). By this time, two additional BDAs had been installed just below the original BDA to create a series of 3 pools intended to facilitate fish passage by jumping over the structures, in addition to the existing passage option via the side channel that flowed around the structure (Fig 2). This second passage option was included because of concerns from regulators (i.e. the California Department of Fish and Wildlife) that upstream moving fish may prefer to jump over structures rather than move around them using a side channel. The original BDA was labeled BDA 1 and the middle and downstream most BDAs were labeled BDA 2 and BDA 3, respectively. To assess whether juvenile salmonids could cross BDAs, on October 24, we captured and tagged 154 juvenile *O. kisutch* and 39 juvenile *O. mykiss* and placed them in the pool below BDA #2 (Fig 2–bottom). Similar to Experiment #1, the portable PIT antennas were placed in the release pool, just above the upper BDA in the pond, and downstream of the block net (Fig 4). Flow during the experimental period is shown in Fig 3.

This arrangement allowed the monitoring of fish use of the release pool, four jumping routes and three side channel passage routes as well as any fish that made it into the lower large BDA pond or other parts of the restoration complex. We were not able to provide complete antenna coverage of all of the lesser flow paths and some fish may have passed undetected by moving through the pores of the BDAs. The temporary antennas were operational for approximately three weeks, from October 24 through November 11, at which point, few fish

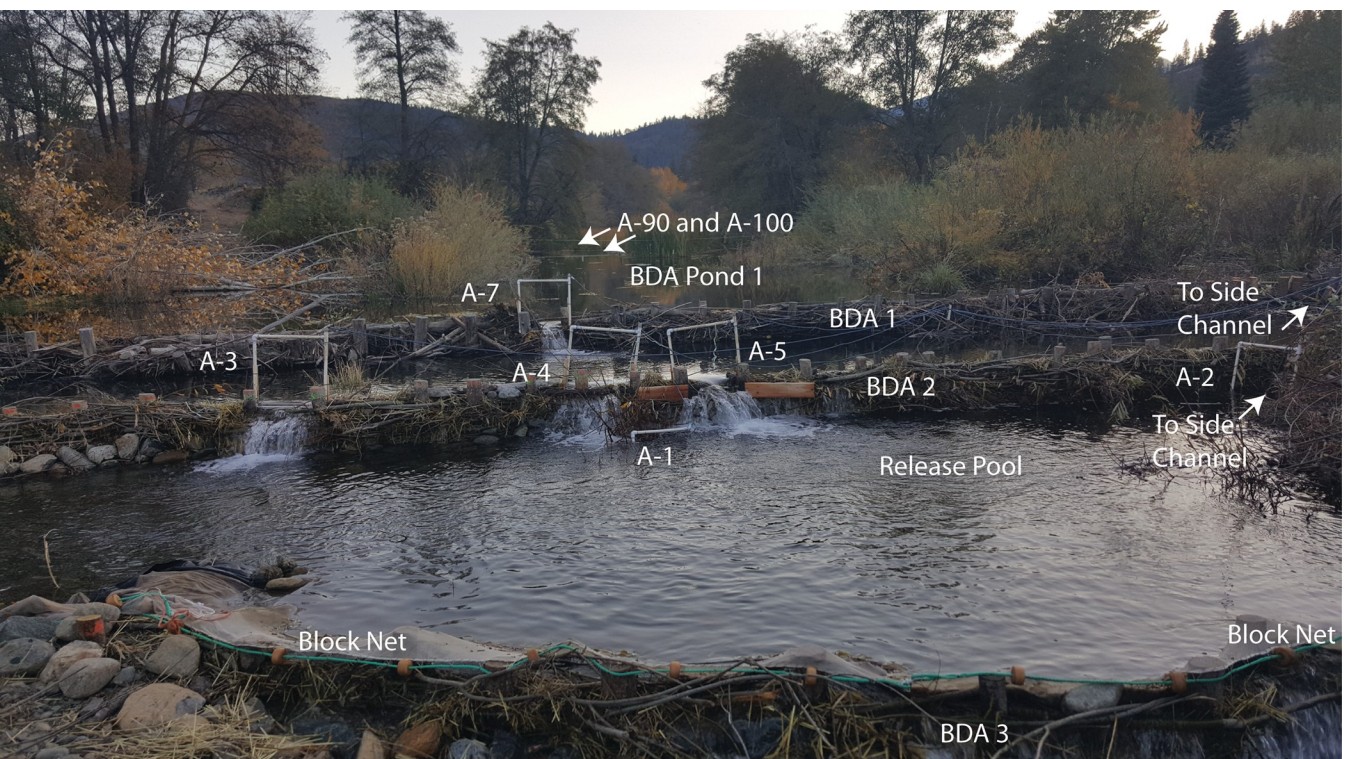

**Fig 4. Experimental layout of the PIT antennas to detect fish passage across BDAs.** Layout of temporary antennas and block net in fall 2017 to monitor the movement of juvenile coho salmon and steelhead trout PIT-tagged and placed in the release pool below BDA 2. Drop over BDA 1 = 27 cm, drop over BDA 2 = 40 cm. Antennas A-3, A-4 and A-5 monitored the three primary jump routes on BDA 2, while Antenna A-7 monitored the single primary jump route available on BDA 1. Antenna A-1 monitored fish in the release pool, and Antenna A-2 was one of 3 antennas that monitored fish passage on the side channels (the other two, A-6 and A-8, are not pictured). Antenna A-90 and Antenna A-100 are approximately (90 and 100 m upstream of the confluence of Sugar Creek and the Scott River, and 30 and 40 m upstream of BDA 1, respectively, and collect data year round. Just out of the picture on the right are antennas on side channels. Also not in view is another antenna below the block net, to detect for any downstream movement past the block net. Note the recently beaver-felled cottonwood (golden leaves) in upper left of photograph.

were detected in either the release pool or the passage routes and the threat of winter storms required removal of the small portable antennas. The larger permanent antennas upstream of the lower BDA (A-90 and A-100) collect data year-round. In 2017, for each major flowpath over or around a BDA, a longitudinal profile of bed and water surface elevation was mapped using a Trimble R8 Model 3 connected to Real-Time Kinematic Global Navigation Satellite System (RTK-GNSS). Velocities were measured at discrete points along the profile and at cross sections to approximate discharge for each of the flow paths, using a SonTek Flowtracker Handheld 2D ADV. Total stream discharge was measured at a California Department of Water Resources monitoring station on Sugar Creek, approximately 700 m upstream from the upper BDA (data available at https://cdec.water.ca.gov/).

There were numerous flow paths over, through and around the BDAs. On BDA 2 (the lower BDA that displaced fish had to cross) major flow paths were a side channel that skirted the edge of the BDA on river left, with a discharge of approximately 0.03 $m^3$/s, and three sections across the top of the BDA, each with a similar amount of discharge, where water flowed over the top to form waterfalls (Fig 4). The side channel flowed for 8 m over cobble and gravel at a slope of 11%, and entered into the pool below BDA 2 (i.e., the "Release Pool", where displaced fish were released). The water surface elevation-to-water surface elevation drop at the falls flowing over the BDA ranged from 38–40 cm.

Major flow paths on BDA 1 (the upper BDA that fish displaced fish had to cross) were a side channel that flowed on the river left side of the main BDA section and a single section near the middle where water flowed over the top of the BDA. Flow out of BDA 1 was much more dispersed, flowed through dense vegetation, and there were numerous passage routes where we were unable to place PIT antennas (Fig 2). The side channel passage route that we were able to monitor flowed over cobble and gravel for a distance of 5 m at an 8% slope, until at the downstream end it entered the pool immediately above BDA 2. The water surface elevation-to-water surface elevation drop at the waterfall over BDA 1 was 27 cm.

**Experiment 3.** During the summer and fall of 2017, we captured and tagged juvenile salmonids in the reach below the BDAs, at the confluence of Sugar Creek with a major side channel of the Scott River (Fig 2). We opportunistically captured and tagged juvenile salmonids on July 25, August 18 and September 19, 2017, for a total of 61 coho salmon, 126 steelhead trout and 12 Chinook salmon tagged during the three events. Flow during those periods is noted in Fig 3. The purpose of this series of experiments was to assess whether fishes naturally summer-rearing in the relatively shallow pool-riffle environment below the BDAs moved upstream and into the pond habitat above the BDAs.

Early fall population estimates upstream of the BDAs were made through a mark-recapture effort on 10/24-10/25/17. Populations of juvenile coho salmon and steelhead trout were estimated using the Lincoln-Petersen mark-recapture method [49]. Juvenile coho salmon habitat capacity upstream of the BDAs was estimated as described in [50, 51]. This method requires measuring the habitat parameters of velocity, depth and proximity to cover and then weighting their value to juvenile coho salmon based on the value of these parameters. We subsampled the habitat for these three metrics along six cross sections within the treated area at approximately equal intervals, then weighted for area based on the actual distance from the mid-points between cross-sections.

All data are available at NOAA's Northwest Fisheries Science Center in Seattle, Washington, USA (https://www.nwfsc.noaa.gov/, UTM coordinates = 10T 552099E, 5277064N).

## Results

### Experimental capture and releases

**Experiment 1.** This initial pilot experiment in 2016 showed that 74% (n = 58) of the juvenile steelhead trout that were placed in the release pool were detected upstream of the single BDA (#1) within 3 days of release, and just a few individuals remained in the release pool 4 days after release (Fig 5). Most of the upstream movement occurred in the hours after sunset, from around 7 pm through midnight, with a smaller pulse near sunrise (Fig 6). There was little upstream movement during the daylight hours, with just 9% (n = 58) fish moving upstream between sunrise and sunset. Three fish exploited small openings in the block net and moved downstream. We also detected below the BDA, three previously tagged fish that were captured and released above the BDA as part of another unrelated study. Additionally, over the course of the 4-day experiment, several tagged fish moved upstream past the BDA, then back downstream and then back upstream again. One fish moved downstream below the block net, then back into the release pool, then upstream above the BDA.

**Experiment 2.** The majority (91%, n = 155) of the tagged juvenile coho salmon that were placed in the release pool below BDA 2 left the pool within 36 hours of being released (Fig 7). The fish were released around 2 pm on October 25th, and by evening of the 26th, just 11% (n = 155) coho salmon remained in the release pool, and a day later, just six remained. By November 8th, at the end of the experimental period, no coho salmon remained in the release pool. Overall, 91% (n = 155) of the fish were eventually detected in BDA Pond 1. Fifty eight

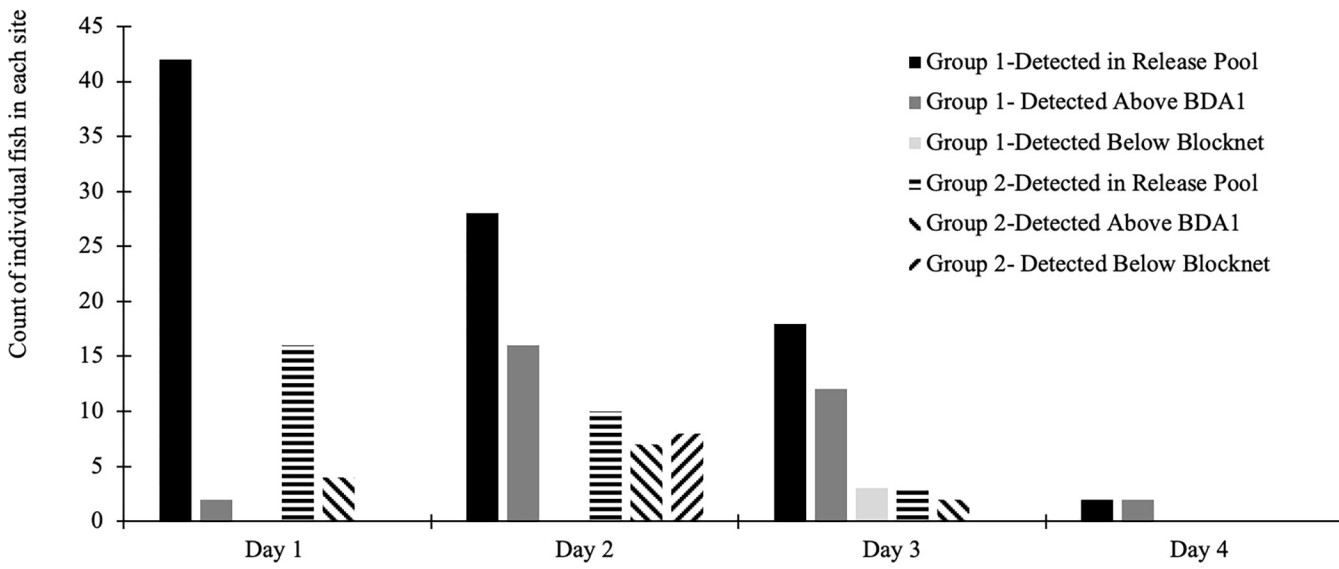

**Fig 5. Daily PIT antenna detections of juvenile steelhead trout above and below BDA in 2016.** The 2016 experimental release of juvenile steelhead trout below BDA 1 on two consecutive days showing the daily number of detections of individual steelhead trout for each release group in the release pool below BDA 1, in the Pond above BDA 1, and below the block net at the lower end of the release pool. Thirty two fish were released in the first cohort on 9/30/16 and 16 more in the second cohort on 10/1/16, for a total of 58 fish released. Day 1 refers to the day of release for each cohort, Day 2, the day after release, etc.

percent (n = 39) of the juvenile steelhead trout also moved up into the beaver pond after release, but not as rapidly as the coho (Fig 8). Eighteen percent (n = 39) of the steelhead were detected in the release pool 60 hours after release, and between five to nine steelhead trout were detected on a daily basis in the release pool throughout the rest of the experimental study

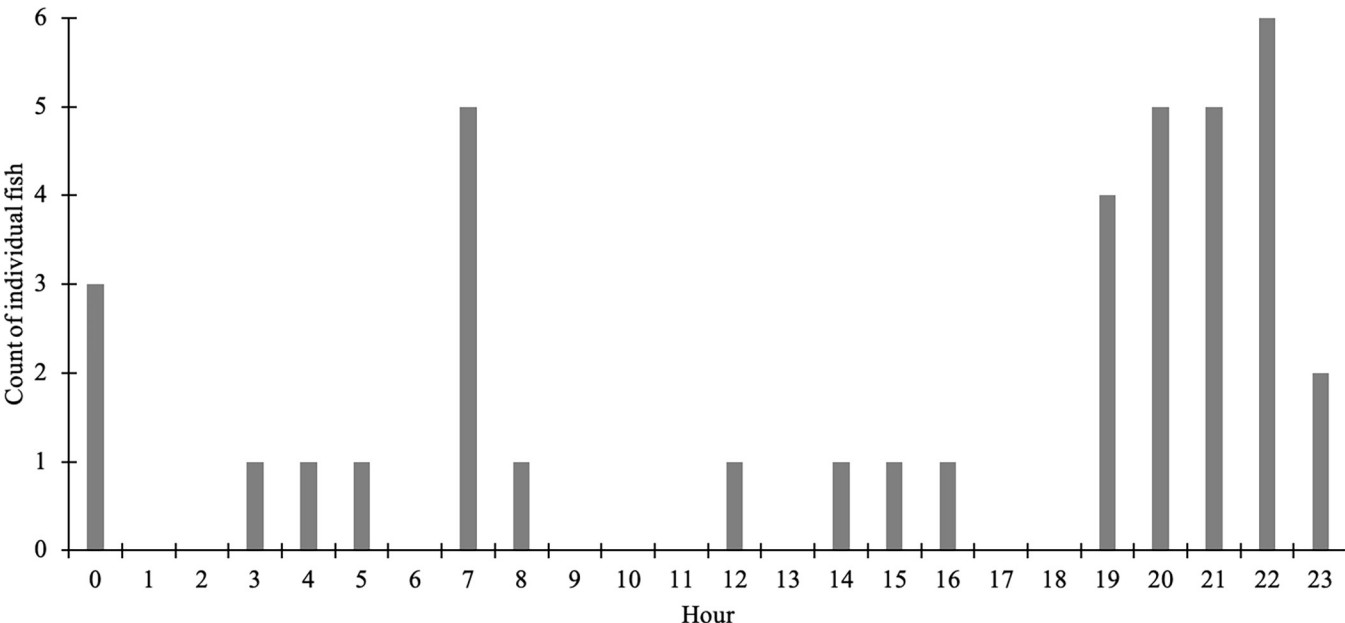

**Fig 6. The hourly timing of movement of juvenile steelhead trout across the BDA in 2016.** The 2016 experimental release, showing the hourly timing of tagged juvenile steelhead trout detected moving past BDA 1 during the 4-day experiment (n = 38 of 58 released). Most of the fish moved during the late evening, after sunset (sunrise = 07:08 and sunset = 18:55 on 9/30/16).

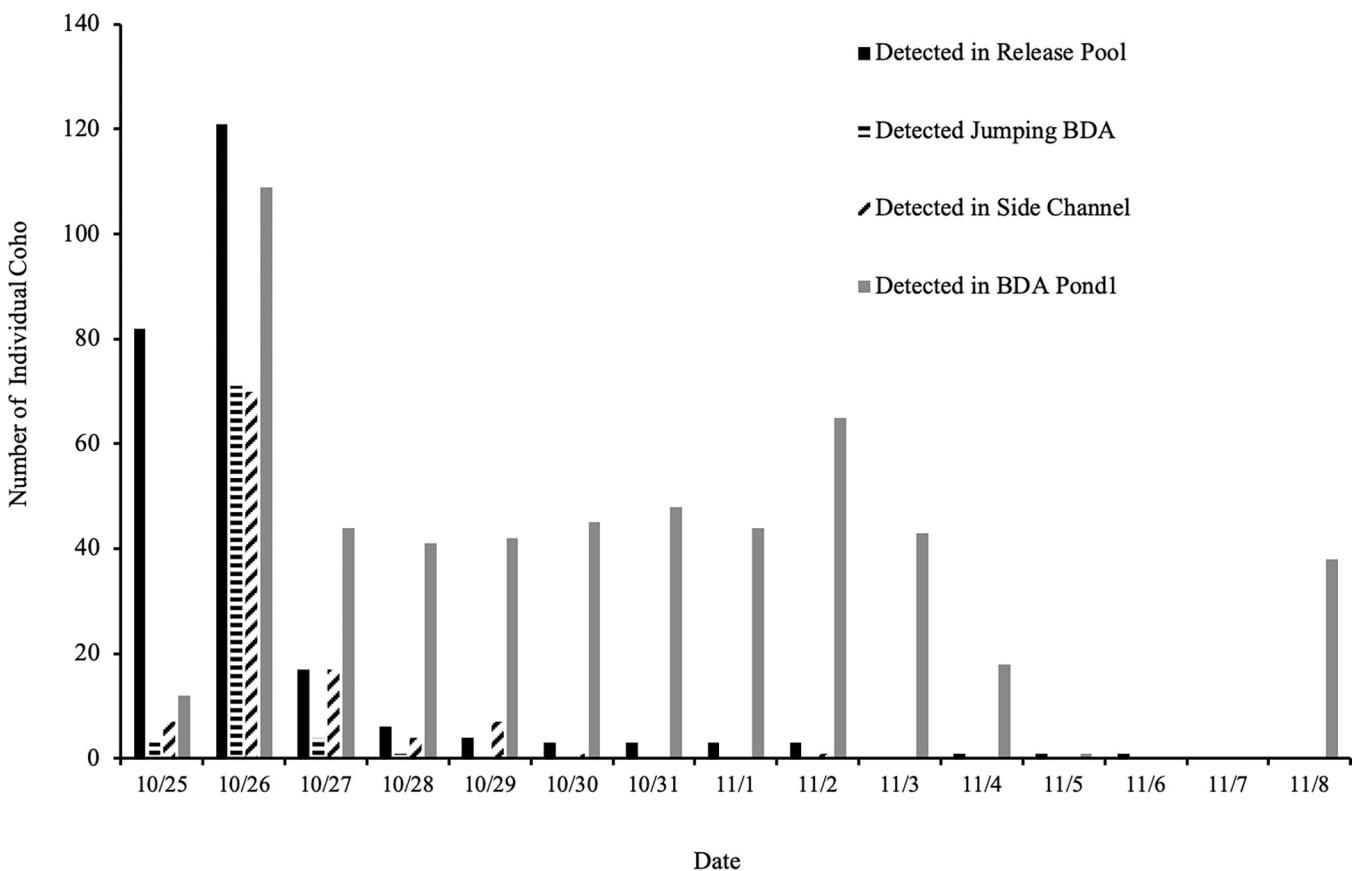

**Fig 7. Daily PIT antenna detections of juvenile coho salmon above and below BDAs in 2017.** Daily detections in 2017 of the number of individual juvenile coho salmon in the release pool, jumping a BDA, using a side channel passage and in BDA Pond 1, and above BDAs 1 and 2 (coho n = 139 of the 155 released). No individuals were detected below the block net on the downstream end of the release pool.

period (through November 8th). Thirteen percent (n = 39) were later detected by one of the permanent antennas in the BDA pond upstream sometime between November 9th 2017, and April 1, 2018, indicating that they had crossed the BDAs.

The percentage of tagged fish detected was high (Table 1). During the initial 21-day study period, all (100%) of the tagged fish were detected at least once somewhere in the antenna network: 93% were detected in the release pool, 94% were detected upstream of the first BDA (BDA 2), and 81% detected upstream of the second BDA (BDA 1). For juvenile coho salmon, 97% were detected upstream of the first BDA (BDA 2), and 90% detected upstream of the second BDA (BDA 1). Overall, a higher percentage of juvenile coho salmon were detected on the upstream PIT antenna network than the steelhead trout (89% v. 50% respectively). No fish were detected on the antenna placed below the stop net to detect any potential downstream escapees.

Sixty percent of the fish used at least one side channel passage to cross a BDA, but many fish chose to jump over at least one of the BDAs (49% for coho, 43% for steelhead), the jump heights of which were 38–40 cm and 27 cm for BDAs 2 and 1, respectively (Fig 9). The lower BDA (2) had three passageways for jumping and of the fish that jumped, there was a strong preference for the river left jump route, for reasons that were not clear. Measurements of velocity profiles and jump heights indicate that the middle and left routes were similar (Fig 9). However 39% of all fish passing BDA 2 used the left jump route, while just 11% used the middle

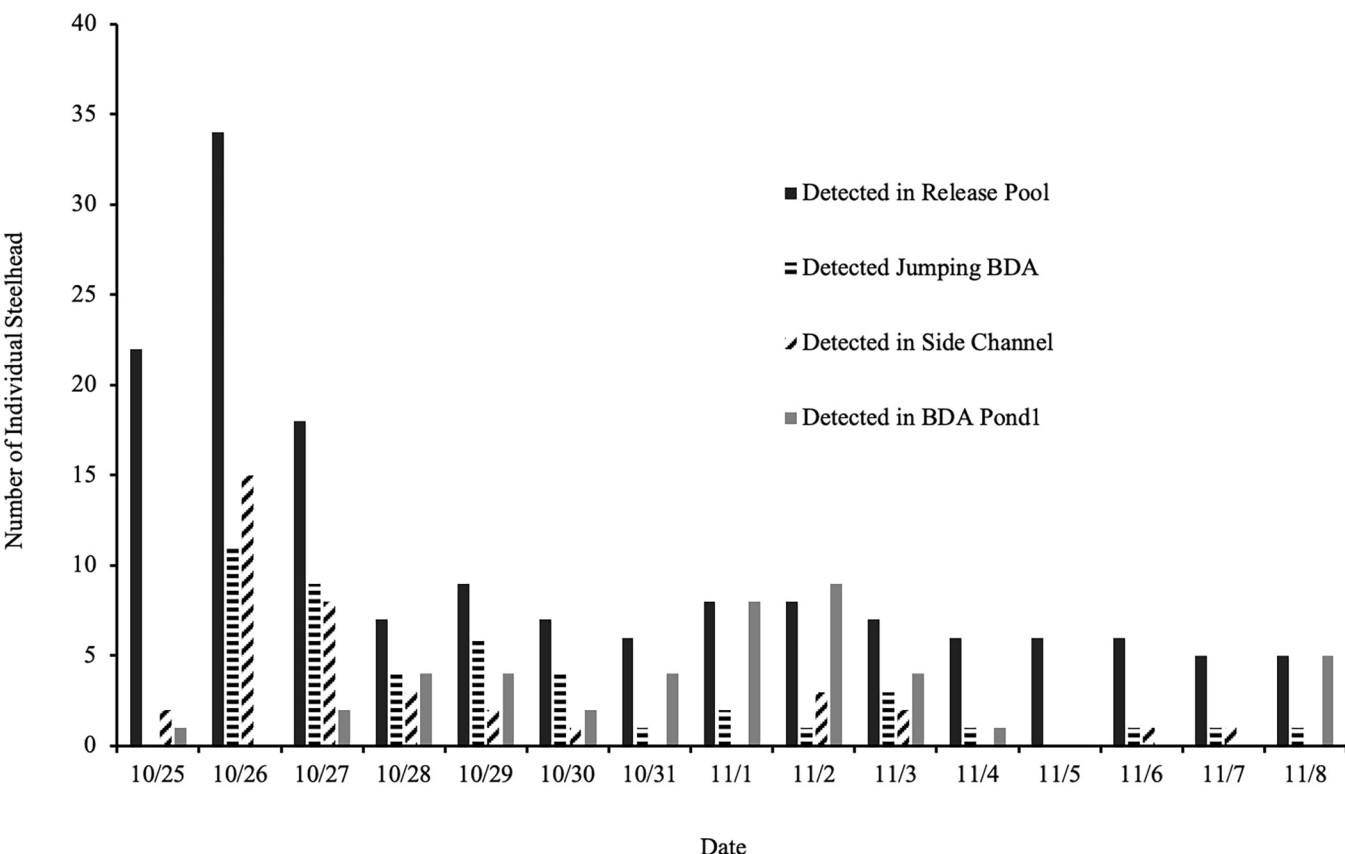

**Fig 8. Daily PIT antenna detections of juvenile steelhead trout above and below BDAs in 2017.** Daily detections in 2017 of the number of individual steelhead trout in the release pool below BDA 2, jumping a BDA, using a side channel passage and in BDA Pond 1, and above BDAs 1 and 2 (n = 39). No individuals were detected below the block net on the downstream end of the release pool.

jump route and just 4% used the right jump route (the remainder used the side channel). The preferred route did have the deepest downstream pool (58 cm), while the right route, which was the least preferred, was in a shallower part of the release pool (23 cm), while the middle jump route had a pool depth of 37 cm. Discharge through the three falls on BDA 2 were approximately equal. Overall, 74/156 (47%) juvenile coho and 17/40 (43%) juvenile steelhead were detected making a 38–40 cm jump over the lower dam.

There were notable behavioral differences between coho and steelhead in terms of the timing of passage and the mode of passage. Of the fish that jumped to cross a barrier, most of the coho jumped between sunrise and sunset, with most of the jumping occurring in the afternoon, while most of the steelhead jumped between sunset and sunrise, with a spike in activity in the hours before sunrise (Fig 10). In contrast, individuals of both species that used the side channel for passage crossed at all hours of the day and night, with spikes in activity for both species at sunrise and the hours after sunset (Fig 11).

**Experiment 3.** 2The fishes captured and released below in the confluence pool in the Scott River (below BDAs 1-3-see Fig 2) in July, August and September, 2017, showed little evidence of upstream movement above any of the BDAs (Table 2). Only 11 of 61 (18%) juvenile coho salmon, 1 of 126 (0.7%) tagged juvenile steelhead trout, and none of the 12 tagged juvenile Chinook salmon were detected upstream of the BDAs through May 31st, 2018. Overall, in total for these three experiments, 187/199 (94%) of the fish tagged in the Scott River, at the confluence with Sugar Creek, were never detected anywhere in the Sugar Creek PIT antenna

**Table 1. Summary of detection and movement of 194 juvenile coho salmon (C) and steelhead trout (S) placed in a release pool (RP) downstream of beaver dam analogues 1 and 2 in October, 2017: SC = Side Channel; L, M and R = left, middle and right, respectively, looking upstream; N = sample size, P = proportion; upper and lower 95% confidence intervals (CI) are in parentheses.**

| Release | Coho | | | Steelhead | | | Total | | |
|---|---|---|---|---|---|---|---|---|---|
| Location | N | P | CI (0.95) | N | P | CI (0.95) | N | P | CI (0.95) |
| All Locations | 155 | 10 | - | 39 | 10 | - | 194 | 10 | - |
| RP | 143 | 0.92 | (0.87–0.96) | 39 | 10 | - | 182 | 0.94 | (0.90–0.96) |
| Above BDA 2 | 152 | 0.98 | (0.94–0.99) | 32 | 0.82 | (0.96–0.92) | 184 | 0.95 | (0.91–0.97) |
| Above BDA 1 | 139 | 0.90 | (0.84–0.94) | 20 | 0.51 | (0.96–0.92) | 159 | 0.82 | (0.76–0.87) |
| Below RP | 0 | 0 | - | 0 | 0 | - | 0 | 0 | - |
| Any BDA Passage | | | | | | | | | |
| Any SC | 93 | 0.60 | (0.52–0.67) | 25 | 0.64 | (0.47–0.79) | 118 | 0.61 | (0.54–0.67) |
| Any Jump | 77 | 0.50 | (0.42–0.57) | 17 | 0.44 | (0.28–0.60) | 94 | 0.48 | (0.42–0.55) |
| BDA 2 Passage | | | | | | | | | |
| L-Jump | 66 | 0.43 | (0.35–0.50) | 11 | 0.28 | (0.15–0.45) | 77 | 0.40 | (0.33–0.47) |
| M-Jump | 8 | 0.05 | (0.03–0.10) | 13 | 0.33 | (0.19–0.50) | 21 | 0.11 | (0.07–0.16) |
| R-Jump | 3 | 0.02 | (0.01–0.06) | 4 | 0.10 | (0.01–0.07) | 7 | 0.04 | (0.02–0.07) |
| All Jumps | 74 | 0.48 | (0.40–0.56) | 17 | 0.44 | (0.28–0.60) | 91 | 0.47 | (0.40–0.54) |
| BDA2 SC | 61 | 0.39 | (0.32–0.47) | 22 | 0.56 | (0.40–0.72) | 83 | 0.43 | (0.36–0.50) |
| BDA 2 Total | 129 | 0.83 | (0.46–0.61) | 31 | 0.79 | (0.64–0.91) | 160 | 0.82 | (0.77–0.87) |
| BDA 1 Passage | | | | | | | | | |
| Jump | 24 | 0.15 | (0.11–0.22) | 0 | 0.00 | - | 24 | 0.12 | (0.08–0.18) |
| SC | 63 | 0.41 | (0.33–0.49) | 15 | 0.38 | (0.23–0.55) | 78 | 0.40 | (0.34–0.47) |
| BDA 1 Total | 83 | 0.54 | (0.46–0.61) | 15 | 0.38 | (0.23–0.55) | 98 | 0.51 | (0.44–0.57) |

network. These results are in contrast to the Experiment 2 fishes that were captured in Pond 1, then released below BDA 2 and then quickly moved back up into Pond 1.

## Population and habitat capacity estimates

We estimated that the amount of habitat created upstream of the BDAs was 7,080 m$^2$ and of a quality sufficient to support 6,744 (SE = 537) coho parr. From our mark-recapture effort, we estimated an actual population of 2,517 (SE = 1173) coho parr, indicating that the habitat was significantly undersaturated. We also estimated the coho survival from summer, 2017 through the 2018 spring outmigration to be 88%. This is based on detection of 863 tagged coho during the spring outmigration period, out of 1077 tagged the previous summer and fall (80.1%), multiplied by an estimated combined PIT antenna probability of detection of 91% for juvenile coho salmon for the two antennas in the lower BDA pond during the spring outmigration period. We did not estimate steelhead abundance because of the relatively low densities observed, but we tagged 361 juveniles in the summer and fall and detected 152 during the spring outmigration period (42%), which multiplied by an 88% antenna probability of detection (for juvenile steelhead trout) provides an overwintering survival estimate of 48%.

## Discussion

To the best of our knowledge, this study provides the first quantitative data for hydraulic conditions (velocity and depth) in the field under which juvenile salmonids cross natural or naturalistic instream barriers such as beaver dams or BDAs both by jumping and through the use of short but steep side channels. This study lends support to the hypothesis that because salmonids have evolved with beaver dams, they have developed behavioral and physical adaptations

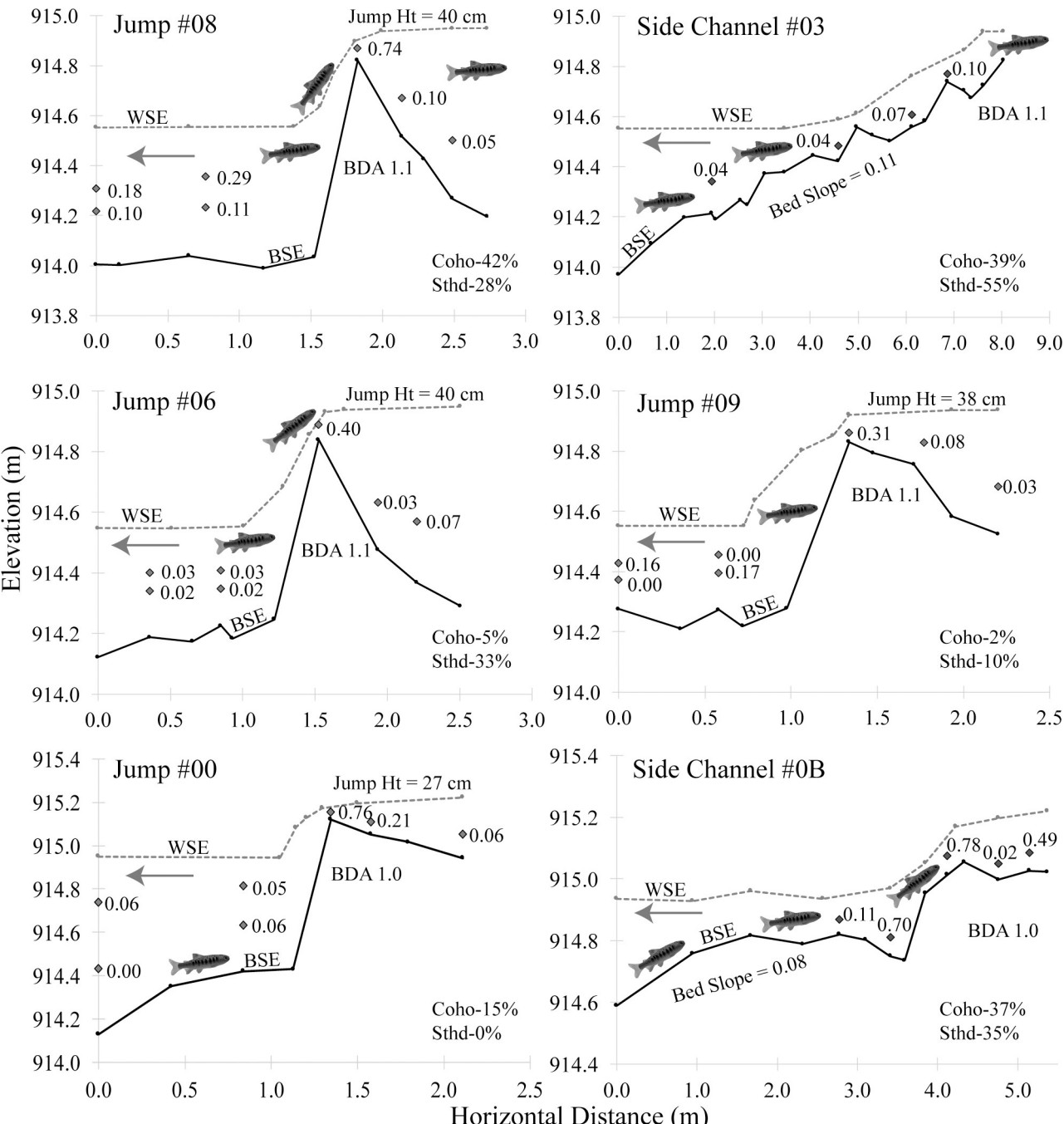

**Fig 9. Examples of the local velocities (m/s) experienced by fish as they followed flow paths to cross the beaver dams, along with the percentage of juvenile coho salmon and juvenile steelhead trout that used that route.** WSE = water surface elevation, BSE = bed surface elevation. Jump Ht = the difference in water surface elevation (cm) between the pool below a dam and the pool above a dam. The locations of each of the pathways are shown in Figs 2 and 4, with the location number referring to the antenna number placed at that location (e.g., Jump #5 is at the location of Antenna A-5). Blue arrows indicate direction of flow.

that allow them to cross such dams at important life-history stages. The two relocation experiments indicate that both coho salmon and steelhead trout parr had little difficulty crossing the BDAs, whether by jumping over a 40-cm waterfall or swimming up a short side channel with an 8–11% slope, the former being somewhat analogous to an engineered pool-weir passage

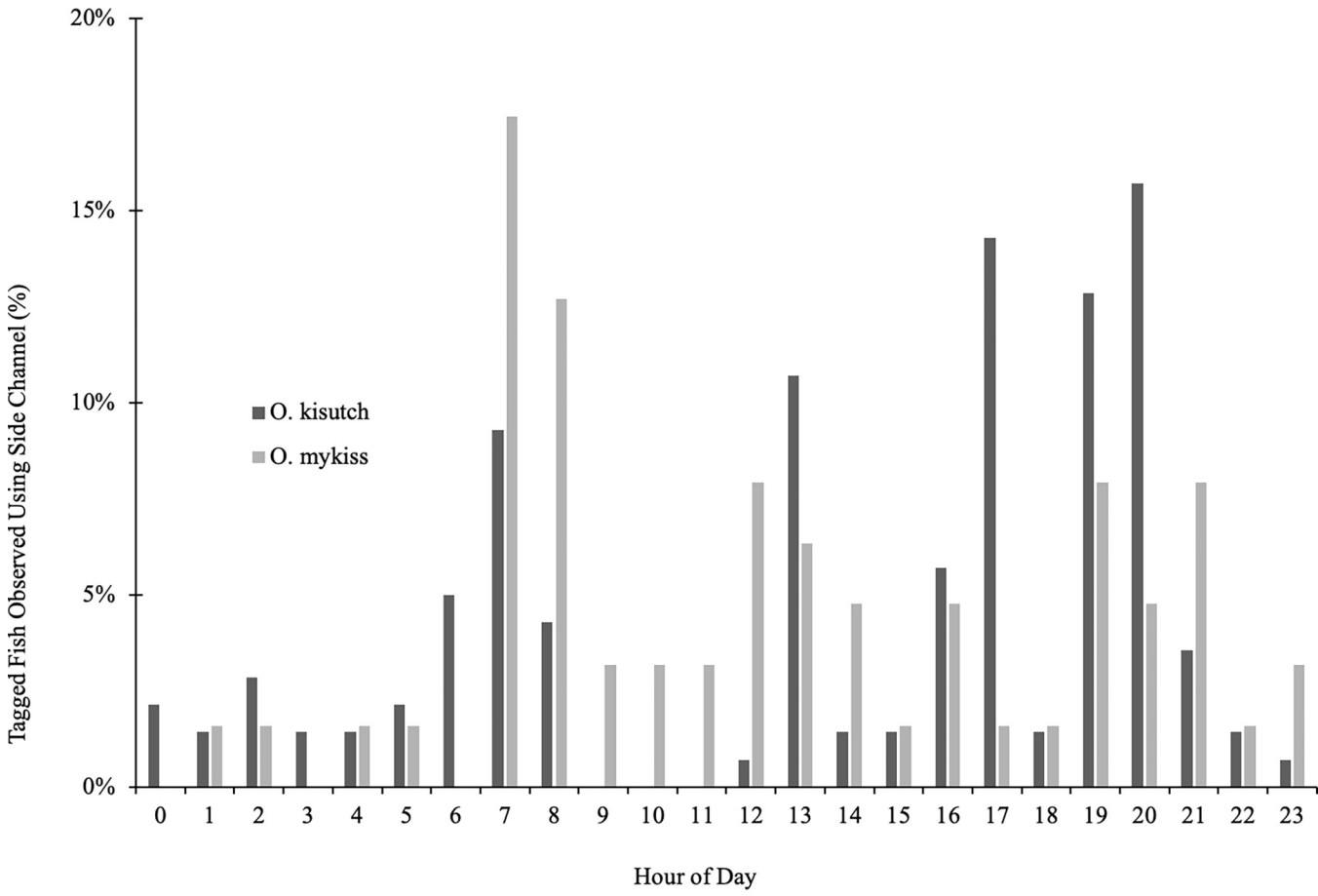

**Fig 10. The hourly timing of juvenile salmonids jumping over BDAs in 2017.** Hourly timing of jumping over BDAs by juvenile coho salmon (*O. kisutch*) and steelhead trout (*O. mykiss*) in 2017, as a percentage of tagged fish (coho n = 155; steelhead n = 39). Coho salmon jumped more frequently during the day, while the majority of steelhead jumping occurred at night. Sunrise = 07:34 and sunset = 18:17 on October 24[th], 2017.

structure and the latter being somewhat analogous to an engineered embedded rock ramp [52]. The fish appeared to time their movements according to light conditions and the majority of them moved upstream within the first or second favorable opportunity. In Experiment #2, a small number of juvenile steelhead trout remained in the release pool throughout the first few weeks of the study, but the majority of those were detected upstream at a later date by the permanent antennas. The upstream antennas detected many more coho salmon relative to steelhead trout, probably in part due to the antenna locations, which were placed in deep slow water habitat favored by coho salmon, as opposed to the faster and more turbulent water preferred by steelhead. We also note that the habitat in the release pool was not poor quality, with good depth, cover and aeration (Figs 4 and 9), and the initial lack of upstream movement by some individuals may have been due to the fact that they found the release pool to be suitable habitat.

There are surprisingly few studies of specific conditions under which juvenile salmonid (or other species) crossed instream barriers, whether natural or artificial, and even fewer studies documenting the hydraulic and hydrologic conditions under which juvenile fishes cross beaver dams, and especially during low-flow conditions. Guidelines for adult fish passage recommend that the pool depth in pool-weir passage routes be at least twice the length of the fish [53, 54], and for ramps that the depth be at least as much as the body height of the fish,

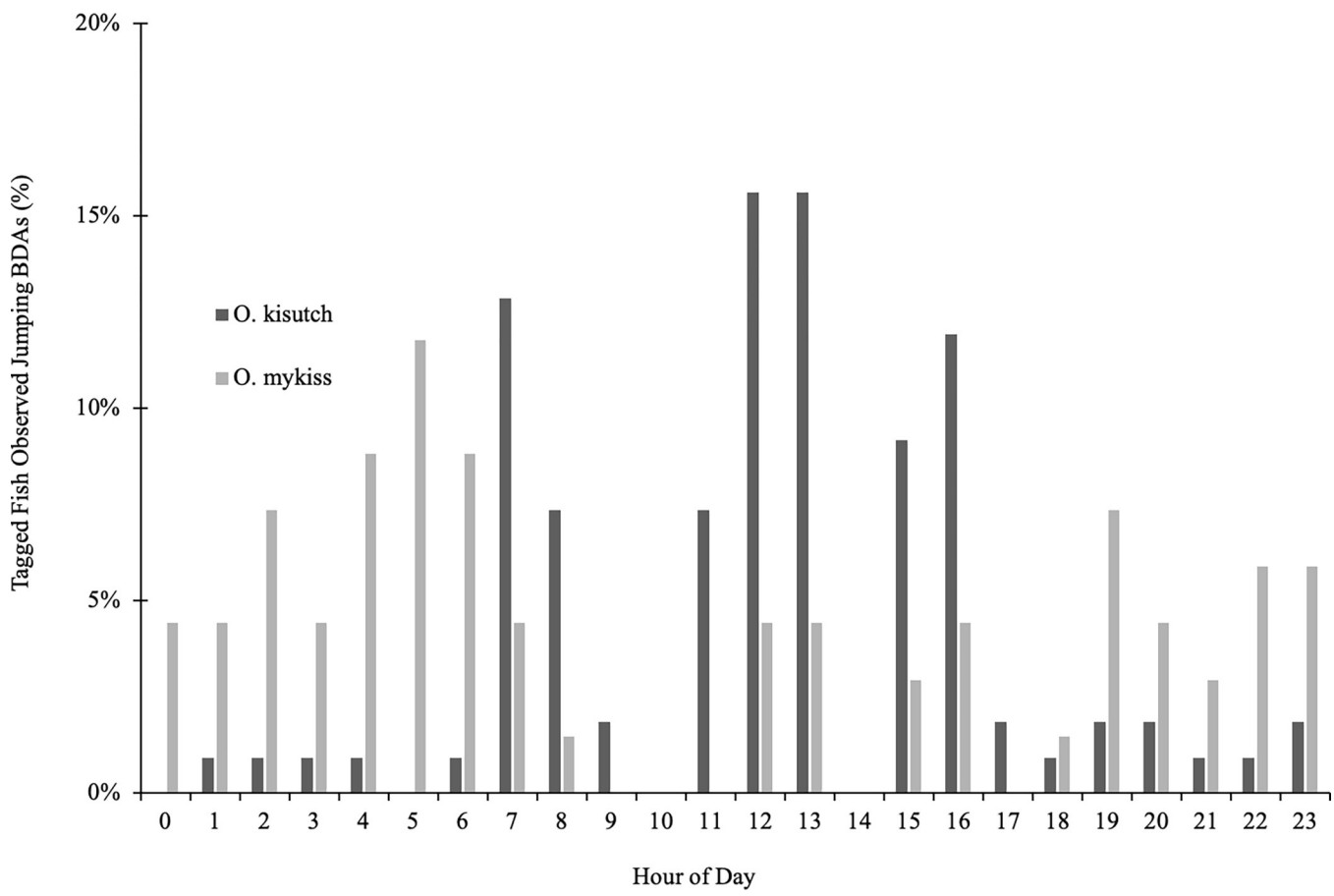

**Fig 11. Hourly timing of side channel usage by juvenile salmonid to pass the BDAs in 2017.** Hourly timing of side channel passage of juvenile coho salmon (*O. kisutch*) and steelhead trout (*O. mykiss*) across BDAs in 2017, as a percentage of tagged fish (coho n = 155; steelhead n = 39). In contrast to jumping, coho preferred side channel passage at night, while the steelhead showed no particular preference between day and night. Sunrise = 07:34 and sunset = 18:17 on October 24[th], 2017.

conditions that were easily met for our juvenile fish in our experimental conditions. Guidelines for jump heights at pool-weir fish passage facilities (i.e. the difference in water surface elevations between two consecutive pools) generally ranges between 15–20 cm [30, 31], conditions that were not met under our "pool-weir" passage option.

That close to half of the juveniles in Experiment #2 crossed a 40-cm (4–5 body lengths) jump, even when a much gentler sloping cobble ramp was available a few meters away indicates that for beaver dams or naturalistic beaver dam-like structures, current jump height guidelines, which generally try to keep jump heights to < 15 cm, may need revising. Such jump height guidelines were generally developed to ensure that fish (primarily salmonids) can pass culverts, diversion dams, hydropower dams and other human-built instream obstructions. However, the hydraulics of such structures are quite different from those of beaver dams. Consider culverts, for example, which fish may have an initial jump to enter, and then face a long swim through high velocity water to reach the upstream end of the culvert. Such a challenge potentially exceeds the swimming capabilities of juvenile or even adult salmonids. In contrast, the hydraulic conditions at the preferred jump passage route at the BDAs require a fish to begin at a pool with a depth of 5–6 body lengths, and then jump or swim up a waterfall that has a high velocity segment extending no more than 15 cm swim distance (about 2 body

**Table 2. Summary of 3 experimental releases of tagged juvenile steelhead trout and coho and Chinook salmon below beaver dam analogues in 2016 and 2017.**

| Species | Total Released | Detected Below BDA 1 N (P, LCL-UCL) | Detected Above BDA 1 N (P, LCL-UCL) | Not Detected Above or Below BDA 1 N (P, LCL-UCL) |
|---|---|---|---|---|
| Expt-1, Sept. 2016: Capture in BDAP1, release in pool below BDA 1 | | | | |
| Steelhead | 58 | 58 (1.00, 0.94–1.00) | 52 (0.90, 0.79–0.95) | 0 (0, 0.00–0.06) |
| Expt-2, Oct., 2017: Capture in BDAP1, release in pool below BDA 2 | | | | |
| Coho | 155 | 152 (0.98, 0.94–0.99) | 141 (0.91, 0.85–0.95) | 0 (0, 0.00–0.02) |
| Steelhead | 39 | 39 (1.00, 0.91–1.00) | 22 (0.56, 0.41–0.71) | 0 (0, 0.00–0.09) |
| Expt-3, July, 2017: Capture and release in confluence pool, no block net | | | | |
| Coho | 12 | 1 (0.08, 0.01–0.35) | 4 (0.33, 0.14–0.71) | 7 (0.58, 0.32–0.81) |
| Steelhead | 1 | 0 (0, 0.00–0.79) | 0 (0, 0.00–0.79) | 1 (1.00, 0.21–1.00) |
| Chinook | 1 | 0 (0, 0.00–0.79) | 0 (0, 0.00–0.79) | 1 (1.00, 0.21–1.00) |
| Expt-3 Aug., 2017: Capture and release in confluence pool, no block net | | | | |
| Coho | 6 | 3 (0.50, 0.19–0.81) | 1 (0.17, 0.03–0.56) | 3 (0.50,0.19–0.81) |
| Steelhead | 15 | 3 (0.20, 0.03–0.56) | 0 (0, 0.00–0.20) | 12 (0.80, 0.55–0.93) |
| Chinook | 5 | 0 (0, 0.00–0.43) | 0 (0, 0.00–0.43) | 5 (1.00, 0.57–1.00) |
| Expt-3, Sept., 2017: Capture and release in Scott-Sugar confluence pool, no block net | | | | |
| Coho | 43 | 19 (0.44, 0.30–0.59) | 6 (0.14, 0.07–0.27) | 22 (0.51, 0.37–0.65) |
| Steelhead | 110 | 13 (0.12, 0.07–0.19) | 1 (0.01, 0.00–0.05) | 96 (0.87, 0.80–0.92) |
| Chinook | 6 | 0 (0, 0.00–0.39) | 0 (0, 0.00–0.39) | 6 (1.00, 0.61–1.00) |

The data are summarized as both the number and proportion of a species detected or not detected above and below BDA 1. See Fig 2 for spatial configuration of the release site relative to BDAs. Total Released = total fish tagged and placed in release pool; N = number detected, P = proportion detected, LCL = lower 95% confidence interval limit, UCL = upper 95% confidence limit.

lengths) followed by a deep ($>$ 6 body lengths), low velocity pool with cover immediately upstream (Fig 9). Thus, not unexpectedly, experiments of fish passage through culverts (usually with hatchery fish) have observed much lower successful jump heights than the results we observed with wild fish jumping over BDAs [55].

In contrast to the jump route, the side channel passage consisted of a short, high gradient (8–11%) flow path, but with extensive channel roughness in the form of cobbles that dissipate energy. For a juvenile salmonid, this flow path appeared as a series of small chutes and pools with the roughness creating turbulent flow and relatively low velocity conditions (Fig 9). These hydraulic conditions are significantly different than a culvert or cement flume angled to a 8–11% slope, which would tend to have more laminar and uniform flow. We did not find specific guidelines for the acceptable length and slope of embedded rock ramps that is thought to ensure fish passage, but recommendations for culverts are generally that the slope should be close to zero or at least consistent with the upstream and downstream stream slope [30, 31]. For adult salmonids, we found examples of sloped flow-ways that provided passage, but the physical characteristics were not well described [56]. For our study, we provided the equivalent of an embedded rock ramp that was generally at least a juvenile fish height deep, with an 8–11% slope, a total distance ranging from about 50–250 juvenile fish lengths and a discharge of about 0.03 m³/s.

## Habitat tradeoffs

Because our study also showed concentrations of juvenile coho salmon and steelhead trout in the ponds upstream of the beaver dams (consistent with other studies), it raises a larger philosophical question as to how to weigh the benefit of the habitat created upstream of a barrier such as a beaver dam against the cost that it might not be passable to all species at all their

different life-history stages and under all flow conditions. Upon review we found that most studies concluded that fishes, and in particular salmonids, benefit from natural obstructions such as beaver dams [6, 26, 38], while studies arguing that beaver dams are detrimental to fish are uncommon, and typically indicate a temporally intermittent negative impact, with no indication of a population-level effect [28].

For example, over a period of 12 years in Nova Scotia, it was observed that in years with low flow, adult Atlantic salmon were unable to pass over some beaver dams and thus spawned lower in the system, but in most years, beaver dams had no detectable effect on the distribution of spawning redds [57]. In Utah it was observed that beaver dams appeared to impede the movement of invasive brown trout (*Salmo trutta*), but not invasive brook trout or native cutthroat trout [29], whereas in California it was observed that brook, brown and rainbow trout regularly crossed beaver dams in both an upstream and downstream direction, but that the loss of beaver dams after severe flooding decreased the brown trout population [58]. In a Midwestern stream it was observed that fish movement of multiple species across beaver dams was linked to flow, with more downstream movement occurring during periods of elevated discharge [59].

None of these studies considered whether there were any population-level effects, nor did they examine similar habitat without beaver as a comparison, or consider that it might be advantageous for fishes not to cross beaver dams. For example, in the Nova Scotia study, it was left undiscussed the possibility that the Atlantic salmon may have found it more advantageous in drought years to spawn in the lower reaches of a stream, and below beaver dams, because of improved flow conditions downstream, potentially a result of water stored behind the upstream beaver dams [57]. As another example, in Washington, a telemetry study of the rare Salish sucker (*Catostomus catostomus*) indicated that they rarely crossed beaver dams, but then a later study at the same site indicated that the highest number of suckers were in the beaver pond complexes, and that the habitat was consistent with habitat descriptions of "good" sucker habitat [60].

At our study site, we demonstrated that the ponds upstream of the BDAs provided summer rearing habitat for thousands of fishes, from a reach that formerly ran dry during the summer. This indicates that breaching the BDAs (and thus draining the ponds) to ensure fish passage would have likely resulted in a net loss of benefit. Because we also demonstrated juvenile fish passage, a decision to breach the BDA (to comply with 15 cm fish passage jump heights) would arguably have been detrimental to the species due to the loss of summer rearing habitat that would have occurred when the pond drained. However, in other situations, where there are not data to assist with decision-making and where flow conditions may be different, the decision of whether to remove or modify an obstruction so that it complies with fish passage guidelines, or to require a proposed restoration structure to comply with fish passage guidelines, may be less clear. The data from our study provide some general guidance, indicating that knowledge of how fish use a particular stream system and the relative abundance of different habitat types within the system is key to understanding how to manage instream obstructions such as beaver dams or BDAs.

We think that because outmigrating juveniles time their downstream movements to coincide with high flows, concerns over passability at this life-history stage are less warranted [61–63]. The same can be said for adult salmon, especially coho salmon, which generally time their movements to coincide with high flows [61, 62, 64]. an assessment of hydraulic conditions at a time when adults are trying to move past the structure is essential to assess whether or not the structure may be blocking movement, but even then, a consideration of the juvenile overwintering habitat that will be lost if the dam is breached needs to be weighed against the potential benefits to having an increased number of fish spawning upstream.

Overall, we suggest that unless there is clear and compelling evidence that a beaver dam or BDAs are preventing the movement of fishes *and* that this is likely to have a population-level effect, such structures should not be removed. Options such as temporarily notching may be an alternative under some conditions, such as the presence of adult salmon stacking up below a dam, but guidelines need developing. For human-built structures such as BDAs and other weirs, we conclude that our data provide some guidance as to what constitutes a passable structure, but that more examples from the field are needed under a wider range of flow conditions.

## Management implications

Studies that assess the costs and benefits of a structure to a fish population are essential, as are studies that continue to assess jump height, route types and among species differences. Because beaver dams and similar structures can provide extensive habitat upstream, the cost of impaired fish passage needs to be weighed against the upstream habitat benefits accrued. In general the benefits of increased connectivity, that is access to habitat, needs to be weighed against the quality of the habitat that is available to use. We speculate that in the case of coho salmon, decades of emphasizing habitat connectivity over habitat quality by removing perceived obstructions to fish passage is a significant contributing factor to their widespread decline.

Studies from Alaska to California have found that where abundant instream obstructions that create deep slow-water habitat, coho salmon thrive, and that conversely, where such habitat is rare or absent, coho salmon are typically rare or absent [45, 65]. While removal of channel-spanning structures that unreasonably restrict the movement of fishes is an important goal, the pursuit of that goal needs to be tempered against the need for creating habitat of a type and quality to which species have adapted. Species have adapted to and evolved in the presence of channel-spanning instream obstructions such as beaver dams and wood jams. Numerous species use the complex and dynamic pool, pond and wetland habitat created by such obstructions and some of those species are in steep population decline. In addition to coho salmon, other regionally rare or endangered species that benefit from beaver ponds include the willow fly catcher (*Empidonax traillii*) and yellow-legged frog (*Rana muscosa*) [66–70]. We find that there is a need for a more nuanced approach to fluvial ecosystem management, an approach that recognizes that a dynamic tension exists between the need for habitat connectivity and habitat quality, between the need for fast-water habitat and slow-water habitat. Management strategies that more explicitly recognize that such variation exists would lead to more successful recovery of a number of aquatic and riparian-dependent species in decline, including coho salmon. In sum, our study suggests beaver dams, BDAs, and other channel spanning habitat features create habitat that benefits numerous species and should be preserved and restored rather than removed as perceived obstructions to fish passage.

## Supporting information

**S1 Data.**
(ZIP)

## Acknowledgments

Numerous individuals contributed to this study in large and small ways and are listed in no particular order of importance. Internal peer-reviewers at NOAA were Colin Nicols, David White and George Pess, all of whom made substantive contributions that improved the quality of this manuscript. Bob Pagliucco of the NOAA Restoration Center provided input into the

study design. Special thanks to Jeff Fowle and the Farmers Ditch Company, LLC and Mike Kalpin for allowing the use of their private property for this experiment. Betsy Stapleton, Charnna Gilmore, Peter Thamer, Wade Dedobbeler, Dale Munson, Linda Baily and Amanda Planck, all of the Scott River Watershed Council contributed to the data collection and restoration effort and the complex logistics of managing an experiment in a remote location. Garreth and Millie Planck of Scott River Ranch, LLC provided early insight and commentary that was helpful in the development of the study design. Mark Cookson of the US Fish and Wildlife Service and Donald Flickinger of NOAA provided initial permitting assistance and project support. We also appreciate the detailed and thoughtful editorial comments of anonymous Reviewer 1.

## Author Contributions

**Conceptualization:** Shari Witmore, Erich Yokel.

**Data curation:** Erich Yokel.

**Formal analysis:** Erich Yokel.

**Investigation:** Erich Yokel.

**Methodology:** Shari Witmore, Erich Yokel.

**Project administration:** Shari Witmore, Erich Yokel.

**Supervision:** Michael M. Pollock, Shari Witmore.

**Writing – original draft:** Michael M. Pollock.

**Writing – review & editing:** Michael M. Pollock, Shari Witmore, Erich Yokel.

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
