## [Decision Letter · Decision Letter 0]

3 Feb 2022

PONE-D-21-27222

A field experiment to assess passage of juvenile salmonids across beaver dams during low flow conditions in a tributary to the Klamath River, California, USA.

PLOS ONE

Dear Dr. Pollock,

Thank you for submitting your manuscript to PLOS ONE. After careful consideration, we feel that it has merit but does not fully meet PLOS ONE’s publication criteria as it currently stands. Therefore, we invite you to submit a revised version of the manuscript that addresses the points raised during the review process.

Re-review of the manuscript indicates most all of the suggested edits asked for were made to the original submission. There are a few additional comments/edits that should be followed in this submission (please see the review). These are minor edits and should not require a substantial amount of time.

We look forward to receiving your revised manuscript.

Kind regards,

Madison Powell, PhD

Academic Editor

PLOS ONE

https://journals.plos.org/plosone/s/file?id=ba62/PLOSOne_formatting_sample_title_authors_affiliations.pdf"

3. We note that [Figures 1 and 2] in your submission contain [map/satellite] images which may be copyrighted. All PLOS content is published under the Creative Commons Attribution License (CC BY 4.0), which means that the manuscript, images, and Supporting Information files will be freely available online, and any third party is permitted to access, download, copy, distribute, and use these materials in any way, even commercially, with proper attribution. For these reasons, we cannot publish previously copyrighted maps or satellite images created using proprietary data, such as Google software (Google Maps, Street View, and Earth). For more information, see our copyright guidelines: http://journals.plos.org/plosone/s/licenses-and-copyright.

a. You may seek permission from the original copyright holder of Figures 1 and 2 to publish the content specifically under the CC BY 4.0 license. 

4. Please upload a copy of Figure 9-11, to which you refer in your manuscript. If the figure is no longer to be included as part of the submission please remove all reference to it within the text.

Additional Editor Comments:

This revision greatly improves upon the original submission. Additional reviews by a previous reviewer and myself concur that a few edit remain (as pointed out in the attached review). These are minor and should not take too much time for the authors to address. The remarks are largely intended to enhance the readability of the manuscript and should be followed.

Reviewers' comments:

Reviewer's Responses to Questions

**Comments to the Author**

1. Is the manuscript technically sound, and do the data support the conclusions?

Reviewer #1: Yes

2. Has the statistical analysis been performed appropriately and rigorously? 

Reviewer #1: Yes

3. Have the authors made all data underlying the findings in their manuscript fully available?

Reviewer #1: Yes

4. Is the manuscript presented in an intelligible fashion and written in standard English?

Reviewer #1: Yes

5. Review Comments to the Author

Reviewer #1: Review of PONE-D-21-27222

A field experiment to assess passage of juvenile salmonids across beaver dams during low flow conditions in a tributary to the Klamath River, California, USA

Pollock, Witmore, Yokel

General Comments

Overall, the revised manuscript is better than the original. I appreciate the authors’conscientious responses to most of my and the other reviewers’comments. A few points remain somewhat shaky, in my opinion. For example, our concerns about the reporting of detection efficiency for the monitoring array are not fully resolved as “efficiency” language persists in the revision. As the authors state in their response letter, estimating detection efficiency was simply not a part of the study: this should be stated in the manuscript. It is not a problem, per se, that this evaluation is missing, but it is a shortcoming that should be acknowledged and perhaps recommended for follow up studies in the discussion. I also still think that the manuscript will be more effective if the authors are more forthright about some of their study intentions and conclusions. For example, the author responses to my original comments #51, 52, 55, 59, and 62 at best only partially addressed the suggestions. I wholeheartedly agree that instream structures generally and beaver dams specifically provide important habitat value and effort should be made to retain and restore such structures. However, if the authors want to convince state and federal management agencies to do a better job restoring such features, I think a more effective narrative will simply spell that out for readers, starting at the beginning of the manuscript and concluding with clear, prescriptive advice at the end. Regarding the latter point, I recommend adding some sub headers to the discussion section, perhaps including an “Implications and Recommendations” or “Improving Habitat” header near the end to sweep up the management talking points the authors wish to communicate to their agency partners.

Specific Comments

1. Title. In their response letter, the authors claim (twice!)to have modified the title to reflect the several study experiments, but that is not the case in my PDF; the title remains the same as the original.

2. Line 20. “The movement of fishes” = “fish movement”

3. Line 41. I think the end of the abstract would still benefit from a statement closer to the heart of the authors’ overall objective: beaver dams, BDA’s, and other channel spanning habitat features should be preserved and restored rather than removed as perceived obstructions to fish passage.

4. Line 73. Ambiguous “this”. Perhaps start the sentence with “Many scientific disciplines” and replace ‘this’ with ‘which’, or similar?

5. Line 76. Another ambiguous “this”.

6. Line 127. Please consider replacing “utilize” with the more efficient synonym “use” here and in other locations throughout the manuscript.

7. Line 131. Add a comma after “that”?

8. Line 227. This is almost a verbatim repeat of the paragraph starting on line 209. The only apparent difference is the mesh size of the seines (1/4” vs 1/8”). Please consolidate.

9. Line 239. I think this is the third sentence about allowable tagging size.

10. Line 251. In their response letter the authors claim to have removed the term “wiggling”, yet it is here and again in line 315.

11. Line 252. “antenna” = “antennas”.

12. Line 286. Please add scientific name for Chinook salmon.

13. Line 349. Some confusion remains regarding experiment numbering. This section includes experiments three, four, and five. Note that the authors response letter states these experiments were renamed 3.1, 3.2, and 3.3. This nomenclature is not in the manuscript.

14. Line 352. Add a comma after 26.

15. Line 456. “Are not entirely clear” = “were not clear”.

16. Line 463. Percentages would be easier to interpret than fractions.

17. Line 492. Verb tense again slipping into present.

18. Line 500. Again, percentages would be easier to interpret.

19. Line 509. I think percentage should be proportion to match data in the table? Looks like a carryover term from the original table.

20. Line 527. These population estimates differ by nearly a factor of three. Any idea why they were so different? The language is also a little clumsy, with the two sentences starting “As of fall, 2017” and “From our fall, 2017”

21. Line 532. Please see previous comments regarding detection efficiency. I think this number is simply the percent of Coho detected above the BDA, an entirely different metric biologically and statistically.

22. Line 536. “apparent survival” is a new term that has not been defined. Furthermore, why is apparent used for steelhead but not for Coho?

23. Line 542. “have” = “had”.

24. Line 550. Another location where efficiency language creeps in inappropriately.

25. Line 554. Habitat choice/occupancy was not really a study element and was not described in the results section.

26. Line 564. The references should follow “body height of the fish”.

27. Line 567. The references should follow “15-20 cm”.

28. Line 575. Add a comma after culverts. “get into” = “enter”.

29. Line 584. “beaver dams” = “BDAs”?

30. Line 585 “Alternatively” = “ In contrast to jump route”.

31. Line 587. Verb tense.

32. Lines 597 – 601. This sentence would be a good fit for the first paragraph of the discussion.

33. Line 602. Might be a good location for a subheader regarding habitat trade-offs. The statement about “dense concentrations of juvenile” salmon and steelhead does not emerge from the abstract or results sections, but it is clearly an important part of the authors’ primary conclusions from the study. Therefore, it should probably be mentioned earlier.

34. Line 615. Might double check the content of reference # 29. I am fairly certain that Brook trout are not native to Utah.

35. Lines 631,644. The authors should check their use of the term dams versus BDA’s, they are used somewhat synonymously in the discussion section.

36. Line 637. Add a comma after however.

37. Lines 647-651. Some citation support would bolster these statements.

38. Line 664. Might be a good location for a subheader about management implications or habitat complexity/conductivity trade-offs, as noted in general comments.

39. Line 664. While cost-benefit studies would be useful, the authors could also make a plug here for better studies related to jump height, route types, and among species differences.

40. Line 681. The flycatcher and frog are presumably regional species, and this should be clarified.

41. Line 687. Consider starting new sentence at the word “Management”.

42. Line 693. “now” = “no”.

6. PLOS authors have the option to publish the peer review history of their article (what does this mean?). If published, this will include your full peer review and any attached files.

Reviewer #1: No

---

## [Author Response · Author response to Decision Letter 0]

31 Mar 2022

PONE-D-21-27-222

Field experiments to assess passage of juvenile salmonids across beaver dams during low flow conditions in a tributary to the Klamath River, California, USA.

Reviewer #1: Review of PONE-D-21-27222

A field experiment to assess passage of juvenile salmonids across beaver dams during low flow conditions in a tributary to the Klamath River, California, USA

Pollock, Witmore, Yokel

General Comments

Overall, the revised manuscript is better than the original. I appreciate the authors’conscientious responses to most of my and the other reviewers’comments. A few points remain somewhat shaky, in my opinion. For example, our concerns about the reporting of detection efficiency for the monitoring array are not fully resolved as “efficiency” language persists in the revision. As the authors state in their response letter, estimating detection efficiency was simply not a part of the study: this should be stated in the manuscript. It is not a problem, per se, that this evaluation is missing, but it is a shortcoming that should be acknowledged and perhaps recommended for follow up studies in the discussion. I also still think that the manuscript will be more effective if the authors are more forthright about some of their study intentions and conclusions. For example, the author responses to my original comments #51, 52, 55, 59, and 62 at best only partially addressed the suggestions. I wholeheartedly agree that instream structures generally and beaver dams specifically provide important habitat value and effort should be made to retain and restore such structures. However, if the authors want to convince state and federal management agencies to do a better job restoring such features, I think a more effective narrative will simply spell that out for readers, starting at the beginning of the manuscript and concluding with clear, prescriptive advice at the end. Regarding the latter point, I recommend adding some sub headers to the discussion section, perhaps including an “Implications and Recommendations” or “Improving Habitat” header near the end to sweep up the management talking points the authors wish to communicate to their agency partners.

Response: Thank you for the very thoughtful general comments. We believe we have addressed these general comments through addressing the specific comments below. We agreed with all of the specific comments and have changed the manuscript to reflect the comments. We genuinely appreciated the time and effort that that reviewer 1 took to edit the manuscript and have included that appreciation in the acknowledgements section

Specific Comments

1. Title. In their response letter, the authors claim (twice!)to have modified the title to reflect the several study experiments, but that is not the case in my PDF; the title remains the same as the original.

We apologize for the error. The title has been changed to “Field experiments to assess passage of juvenile salmonids across beaver dams during low flow conditions in a tributary to the Klamath River, California, USA.”

2. Line 20. “The movement of fishes” = “fish movement”

Response: We agree and have made the requested change.

3. Line 41. I think the end of the abstract would still benefit from a statement closer to the heart of the authors’ overall objective: beaver dams, BDA’s, and other channel spanning habitat features should be preserved and restored rather than removed as perceived obstructions to fish passage.

Response: We agree, that is a good summary to place at the end of the abstract and have included the suggested language.

4. Line 73. Ambiguous “this”. Perhaps start the sentence with “Many scientific disciplines” and replace ‘this’ with ‘which’, or similar?

Response: The last two sentences of this paragraph were a bit clumsy and a bit repetitive. We have simplified the language to read as follows: Many scientific disciplines related to the study of rivers such as ecology, geology, and fluvial geomorphology emerged in the late 19th and early 20th centuries, and subsequent to the widespread removal of these obstructions to flow and sediment transport., This has profoundly influenced the perception among scientists and natural resource managers, even to this day, that the natural and ideal condition of all streams is “free-flowing” and clear of dams and other obstructions [2, 10, 15]. 

5. Line 76. Another ambiguous “this”.

Response: Addressed in our response to comment #4 above.

6. Line 127. Please consider replacing “utilize” with the more efficient synonym “use” here and in other locations throughout the manuscript.

Response: We agree and have changed “use” to “utilize” throughout the manuscript.

7. Line 131. Add a comma after “that”?

Response: We have added a comma as requested.

8. Line 227. This is almost a verbatim repeat of the paragraph starting on line 209. The only apparent difference is the mesh size of the seines (1/4” vs 1/8”). Please consolidate.

Response: We agree and have consolidated the paragraphs.

9. Line 239. I think this is the third sentence about allowable tagging size.

Response: We agree and have mentioned the tagging size only once.

10. Line 251. In their response letter the authors claim to have removed the term “wiggling”, yet it is here and again in line 315.

Response: We agree and have removed the term wiggling and replaced with moving.

11. Line 252. “antenna” = “antennas”.

Response: We agree and have made the requested change.

12. Line 286. Please add scientific name for Chinook salmon.

Response: We agree and have made the requested change.

13. Line 349. Some confusion remains regarding experiment numbering. This section includes experiments three, four, and five. Note that the authors response letter states these experiments were renamed 3.1, 3.2, and 3.3. This nomenclature is not in the manuscript.

Response: We agree and have removed reference to 3.1, 3.2 and 3.3. It was not necessary.

14. Line 352. Add a comma after 26.

Response: We agree and have added a comma.

15. Line 456. “Are not entirely clear” = “were not clear”.

Response: We agree and have made the requested change.

16. Line 463. Percentages would be easier to interpret than fractions.

Response: We agree and have included percentages.

17. Line 492. Verb tense again slipping into present.

Response: We agree and have change this to the past tense.

18. Line 500. Again, percentages would be easier to interpret.

Response: We agree and have included percentages.

19. Line 509. I think percentage should be proportion to match data in the table? Looks like a carryover term from the original table.

Response: We agree and have changed percentage to proportion.

20. Line 527. These population estimates differ by nearly a factor of three. Any idea why they were so different? The language is also a little clumsy, with the two sentences starting “As of fall, 2017” and “From our fall, 2017”

Response: We agree the language is a bit clumsy and have cleaned it up. Regarding the numbers, one is a habitat capacity estimate and one is a population estimate, which we have clarified. What this shows is that the number of fish using the habitat is significantly undersaturated relative to the capacity of the habitat to support fish.

21. Line 532. Please see previous comments regarding detection efficiency. I think this number is simply the percent of Coho detected above the BDA, an entirely different metric biologically and statistically.

Response: We think that probability of detect is a less confusing term than detection efficiency and have made the language change. The percentage of coho detected above the BDA detected was 80.1%, but this needs to be multiplied by the probability of detection by the antennas, which is not 100% but 91%.

22. Line 536. “apparent survival” is a new term that has not been defined. Furthermore, why is apparent used for steelhead but not for Coho?

Response: We agree and have removed the term “apparent” so that there is consistency between steelhead trout and coho salmon.

23. Line 542. “have” = “had”.

Response: We agree and have change this to the past tense.

24. Line 550. Another location where efficiency language creeps in inappropriately.

Response: We agree and have removed the efficiency language.

25. Line 554. Habitat choice/occupancy was not really a study element and was not described in the results section.

Response: We agree and have removed the language about habitat choice.

26. Line 564. The references should follow “body height of the fish”.

Response: We agree and have placed the references where requested.

27. Line 567. The references should follow “15-20 cm”.

Response: We agree and have placed the references where requested.

28. Line 575. Add a comma after culverts. “get into” = “enter”.

Response: We agree and have made the requested changes.

29. Line 584. “beaver dams” = “BDAs”?

Response: We agree and have changed beaver dams to BDAs.

30. Line 585 “Alternatively” = “ In contrast to jump route”.

Response: We agree and have made the requested changes.

31. Line 587. Verb tense.

Response: We agree and changed appears to appeared

32. Lines 597 – 601. This sentence would be a good fit for the first paragraph of the discussion.

Response: We agree and have placed this in the first paragraph of the discussion.

33. Line 602. Might be a good location for a subheader regarding habitat trade-offs. The statement about “dense concentrations of juvenile” salmon and steelhead does not emerge from the abstract or results sections, but it is clearly an important part of the authors’ primary conclusions from the study. Therefore, it should probably be mentioned earlier.

Response: We agree and have created a sub-header “habitat trade-offs.

34. Line 615. Might double check the content of reference # 29. I am fairly certain that Brook trout are not native to Utah.

Response: The reviewer is correct and we have made the change to reflect that Brook trout are not native to Utah.

35. Lines 631,644. The authors should check their use of the term dams versus BDA’s, they are used somewhat synonymously in the discussion section.

We agree and have changed dams to BDA or BDAs, as appropriate.

36. Line 637. Add a comma after however.

Response: We agree and have made the requested change.

37. Lines 647-651. Some citation support would bolster these statements.

Response: We agree and have added some references in support of these statements.

38. Line 664. Might be a good location for a subheader about management implications or habitat complexity/conductivity trade-offs, as noted in general comments.

We agree and have added a subheader “Management implications”.

39. Line 664. While cost-benefit studies would be useful, the authors could also make a plug here for better studies related to jump height, route types, and among species differences.

Response: We agree and have suggested that additional studies related to jump height, route types, and among species differences would be helpful.

40. Line 681. The flycatcher and frog are presumably regional species, and this should be clarified. 

Response: We agree and have clarified that these are regional species.

41. Line 687. Consider starting new sentence at the word “Management”.

Response: We agree and have started the sentence with the word “Management”.

42. Line 693. “now” = “no”.

Response: We agree and have made the requested change.

---

## [Editor Report · Decision Letter 1]

22 Apr 2022

Field experiments to assess passage of juvenile salmonids across beaver dams during low flow conditions in a tributary to the Klamath River, California, USA.

PONE-D-21-27222R1

Dear Dr. Pollock,

We’re pleased to inform you that your manuscript has been judged scientifically suitable for publication and will be formally accepted for publication once it meets all outstanding technical requirements.

Kind regards,

Madison Powell, PhD

Academic Editor

PLOS ONE
---

## [Editor Report · Acceptance letter]

6 May 2022

PONE-D-21-27222R1 

Field experiments to assess passage of juvenile salmonids across beaver dams during low flow conditions in a tributary to the Klamath River, California, USA. 

Dear Dr. Pollock:

I'm pleased to inform you that your manuscript has been deemed suitable for publication in PLOS ONE. Congratulations! Your manuscript is now with our production department. 

Kind regards, 

on behalf of

Dr. Madison Powell 

Academic Editor

PLOS ONE